# Developing a Business Plan for a Library Publishing Program

Kate McCready  and Emma Molls *

University Libraries, University of Minnesota, Minneapolis, MN 55455, USA; mccre008@umn.edu
*   Correspondence: emolls@umn.edu; Tel.: +1-612-626-5218

**Abstract:** Over the last twenty years, library publishing has emerged in higher education as a new class of publisher. Conceived as a response to commercial publishing practices that have strained library budgets and prevented scholars from openly licensing and sharing their works, library publishing is both a local service program and a broader movement to disrupt the current scholarly publishing arena. It is growing both in numbers of publishers and numbers of works produced. The commercial publishing framework which determines the viability of monetizing a product is not necessarily applicable for library publishers who exist as a common good to address the needs of their academic communities. Like any business venture, however, library publishers must develop a clear service model and business plan in order to create shared expectations for funding streams, quality markers, as well as technical and staff capacity. As the field is maturing from experimental projects to full programs, library publishers are formalizing their offerings and limitations. The anatomy of a library publishing business plan is presented and includes the principles of the program, scope of services, and staffing requirements. Other aspects include production policies, financial structures, and measures of success.

**Keywords:** business plan; publishing; academic libraries; open access

## 1. Introduction

Academic publishing, fueled by the boom of digital internet technologies, has created space for new types of publishers, including library as publisher. Because of the growth of new library publishing programs, and the distinctiveness of scholarly communication approaches across institutions, this paper advocates for the creation and adoption of business plans within library publishing programs. The foundations of library publishing are presented, along with examples of current library publishing programs. This paper walks through a business plan template that can be used by current and future library publishers. Readers working in established library publishing programs that currently lack a business plan, and readers who are considering launching a library publishing program, will find a number of guiding questions for each section of the included business plan template. Finally, the authors hope that this paper engages the entire library publishing community and increases the number of publicly available library publishing business plans.

## 2. Development of Library Publishing Programs

Academic library publishing programs first saw adoption in the early 2000s and have continued to grow over the last two decades [1]. Since 2014, the Library Publishing Coalition (LPC), a membership organization made up of mostly North American academic libraries, has increased membership by over 35%, and has surveyed 125 academic libraries who identify as actively engaging in publishing. Thirty-six percent of LPC members established their publishing program in the last ten years [2].

Beyond the Library Publishing Coalition, " . . . most of the 123 ARL (Association of Research Libraries) member libraries are engaged in publishing or publishing support activities" [3]. Libraries started publishing programs for a variety of reasons, including "mission-aligned work for exploring new opportunities in the digital age [ . . . ], demonstrating the market for scholarly, peer-reviewed, open access monographs, and empowering the library to engage with and effect changes in scholarly publishing" [3]. Although the goals of individual programs may vary, overall, library publishing programs " . . . are focusing on the capabilities and possibilities of new models" and working to avoid the "replicat[ion of] traditional publishing services" [4].

For many libraries, providing publishing services is an extension of a larger suite of scholarly communication offerings, offered frequently to " . . . advance a strategic objective of transitioning the library's collecting activities away from licensing content and towards supporting open access to scholarship" [5]. Although libraries addressing scholarly communication issues was discussed as far back as 1979, scholarly communication service efforts vary greatly across libraries [1]. In a 2015 survey, Ithaka S+R found that across 10 surveyed institutions, scholarly communication programs rarely share organizational structures, functions, and objectives [5]. The varying makeup of scholarly communication programs, combined with the relatively new, and often experimental nature, of library publishing services, leaves libraries to newly navigate the complex landscape of open access publishing.

Unlike other scholarly communication services within a library where the costs have been typically been absorbed by assigning new duties to existing staff, or hiring new staff with new skill sets, the expenditures made on behalf of publishing activities require creative thinking to ensure that necessary elements transform a document into a publication (e.g., having a reputable authority vet the content, applying production techniques to the content, making the published work available through distribution networks, etc.). These early days of library publishing are seeing an examination of which elements that go into creating a publication are necessary to instill trust and produce high quality scholarship while also examining how those activities should be paid for. Business planning for library publishing examines both of these elements.

*Open Access Context: Library Publishing as Disruption*

Most library publishers firmly align with the open access movement which " . . . had its origins in the crisis in scholarly communication and publishing, which has both caused and is the result of declining collections budgets, more demand for newer, expensive resources, and greatly increased pricing for serials, electronic resources, and other library materials [6]". As of 2018, 82% of library publishing programs focused entirely or almost entirely on open access publications [2]. The Budapest Open Access Initiative' Berlin Declaration on Open Access to Knowledge in the Sciences and Humanities focuses on scholarly publishing's results: to make the knowledge created and published open for reading and reuse. The process of getting there is less straightforward. Some institutions and individual authors attempt to achieve open access through the piecemeal deposit of a copy of the work in an institutional repository. Others rely on author processing fees (or APCs) to create an open copy of the published work. But there is a finite amount of money within scholarly publishing. Expenditures on these "solutions" are not relieving the pressure on library collection budgets. In the 2017 *Monitoring the Transition to Open Access* report focused on the UK, the findings (based on a sample of 10 UK universities) suggest that subscription expenditures have grown 20% since 2013 (or an increase of £3 million) while APC expenditures for those institutions grew from £750,000 to £3.4 million. In 4 years, those 10 institutions spent an additional £5.6 million while at the time of publication, 63% of materials remain locked behind a paywall [7]. A growing number of libraries are now asking, can library budgets support the production of scholarly publications differently? Can they instead support the production in a new system where they know and control the costs? Furthermore, current conversations ask, can academia achieve its open access aspirations while continuing to support the commercial models of production [8]?

Although library publishers make up only a tiny fraction of the scholarly publishers in existence, they are attempting to shift the ecosystem. Instead of spending library resources to purchase bundled collections of titles where subscription and production costs are hidden, some institutions are applying a portion of those resources to the production and publication of those works. Libraries are allocating funding to support infrastructure for launching new publications that may not have fit into the legacy commercial publishing model. Charles Watkinson writes "If visualized as a spectrum from informal to formal, the formal book (or journal) occupies a narrow space at the right-hand end of the continuum. To its left lie the many other types of publishing and dissemination needs that a campus community may have" [9].

In 2012, the research report "Library Publishing Services: Strategies for Success" noted that "The vast majority of library publishing programs (almost 90%) were launched in order to contribute to change in the scholarly publishing system, supplemented by a variety of other mission-related motivations. The prevalence of mission-driven rationale aligns with the funding sources reported for library publishing programs, including library budget reallocations (97%), temporary funding from the institution (67%), and grant support (57%). However, many respondents expect a greater percentage of future publishing program funding to come from service fees, product revenue, charge-backs, royalties, and other program-generated income" [10]. It is questionable if it is in the best interest of scholarly communications to attempt to continue supporting, or adopting, the business models used by commercial publishers. Libraries are hiring staff, and engaging with third party vendors, to support publishing services that are grounded in providing both technology support for publishing software systems and production services. They are learning about the necessary production work and finding expertise outside of the library to perform required tasks that aren't typically available within a library's staff's skillset. Importantly, they do not necessarily need to recoup those costs; however, they must spend those dollars judiciously and produce knowledge resources that benefit both their campus and the broader scholarly publishing landscape. Therefore, they need a wholly new business model that holds them accountable to high quality standards, and fulfills their mission, while also being fiscally responsible agents of the dollars entrusted to them.

## 3. Institutional Budget Models and Their Impacts

Libraries, and U.S.-based academic libraries in particular, typically receive the majority of their funding from state appropriations, tuition, and grant awards. Based on data collected by the Association of Research Libraries, 90% of public university library budgets are from state or institutional allocations [11]. The type of budget model used at an institution, and how that model determines the process by which money is allocated to units, will likely have an impact on the services offered by the library. Some budget models have disincentives for attempting cost-recovery for operations while others make it politically difficult to serve "clients" that are not directly affiliated with the university. Additionally, an institution's budget model can have an impact on how library publishing services are funded. A variety of different budget models used in institutions of higher education are explained very well in *Budgets and Financial Management in Higher Education* by Margaret J. Barr, George S. McClellan. As listed in column 1 of Table 1, the book's authors detail the types of structures used at institutions of higher education. Examining these structures, the authors of this article outline some potential impacts on starting or funding a library publishing service in column 2.

**Table 1.** Higher Education Institution Budget Models.

| Type of Budget Models (This column based on, *Budgets and Financial Management in Higher Education* [12]) | Potential Effect on Library Publishing Programs |
|---|---|
| **All Funds**—Emphasizes a holistic goals-oriented perspective. Takes into account all sources of revenue and expense. Facilitates the monitoring of resource allocation in pursuit of institutional goals. | May need library publishing to be seen as an institutional goal, or there is a related goal of transforming scholarly publishing. Cost-recovery income may be considered "revenue" that is scooped. |
| **Formula**—Relies on the use of specified criteria in allocating resources. Development of the formula is critically important. Retrospective in nature. | Formulas are typically developed at a very high level (based on enrollments, or facilities costs) so the overall library budget could fluctuate. Cost recovery may be difficult if units cannot keep their own income. |
| **Performance Based**—Allocation of resources premised on attainment of performance measures. Strength in linking state priorities for higher education to resource allocation. | Performance measures are often tied to graduation or job placement rates. Library publishing may not be seen as contributing to those performance measures. Cost recovery may be difficult if units cannot keep their own income. |
| **Incremental**—Establishes across the board percentage changes in expenditures over current budget based on assumptions regarding revenues for coming year. | Assuming the library is allowed to reallocate funds internally, this would allow for the development and growth of library publishing. Cost recovery revenues may affect future allocations. |
| **Initiative-Based**—Requires units to return portion of their budgets for the purposes of funding new initiatives. Units apply to the pool to support new initiatives. | Requires successful application to begin, or grow services. Growth may need to be self-funded through cost-recovery activities if the initiative funding is one-time vs. recurring. |
| **Planning, Programming, and Budgeting Systems**—Premises on tightly integrating strategic planning, budgeting, and assessment. Decisions are a function of identified challenges and opportunities, weighing risk/reward ratios, and monitoring performance. | Similar to Performance Based. Cost-recovery income may difficult if units cannot keep their own income. Requires a great deal of planning and staff to calculate and monitor the work. |
| **Responsibility Center**—Locates responsibility for unit budget performance at the local level. Units are seen as revenue centers or cost centers. Units are allowed to retain some portion of end-of-year budget surplus. | Other revenue-generating units are "taxed" for library services making it difficult to do additional cost-recovery. Increased scrutiny on serving externally-owned publications which may require complete cost recovery when serving societies/non-profits. More likely that the program/library would be able to keep cost recovery revenues in their own budget. |
| **Zero Based**—Each item in the budget must be justified at the time the budget is developed. Assures active monitoring of the link between institutional activities and institutional goals. | Library publishing must be a goal of the institution. Requires a great deal of staff effort each year to justify the programs' existence. |

Prior to determining the scope of service, or the financial structure of the publishing program, questions about the institution's budget model that should be asked include:

- Does the institutions' budget model prevent cost-recovery activities?
- If costs are recovered, and revenue is generated, does that money need to be given back to the university?
- Are allocated or revenue generated funds scooped at the end of the year (i.e., spend or return to the university)?
- Can the library's publishing unit support external publications? Or, for political reasons, does there need to be a university affiliation with the publication?
- Does the university recognize the benefits of library publishing? What case needs to be made that library publishers are necessary, effective disruptors to the current scholarly publishing environment?
- How can library publishing get an initial allocation? Can it be done at the library level or the university level?

- How can library publishing tie its goals to that of the institution's? Does the university have a mission to support the public (e.g., land-grant mission)?

*Content Creation as Service*

The financial framework in which libraries operate is important to explore before attempting to determine the aspects of a library publishing business plan. Libraries at academic institutions are considered to be a common good. They allocate substantial resources to building collections through traditional collection development activities in order to provide content to users without charge. Libraries typically have missions that aim to provide access to content to all patrons free from barriers. Egalitarian, justice-oriented principles prevail throughout their value statements and are expressed thoroughly in the American Library Association's Core Values [13]. By their nature and their primary aim, libraries strive to get the information that is needed or wanted into a patron's hands as quickly and barrier-free as possible regardless of who that person is or what they want to do with the information.

Academic Libraries may recoup some of their costs, fine patrons for late, damaged, and lost books, or generate income on services such as outward facing research or document delivery services; however, there are no examples of those charges or services fully supporting the primary mission of collecting and delivering resources. As Quinn and Innerd write in their analysis of the integration of their university press into the library: " . . . the library operates under a budget-allocation model provided entirely by the university . . . the centrality of the library to the teaching and research mission of the university is generally accepted and understood. The library's budget has traditionally been based on historical spending and the ability of the library to articulate its need for additional funding to innovate and meet student and faculty demands. The library's goal is to spend wisely, efficiently, and as fully as possible within the budget provided" [14].

This philosophy and approach applies to nearly all scholarly communication oriented services provided by academic libraries: data curation and management, digital scholarship support, institutional repository services, digital library development, research consultations, etc. This prevailing philosophy and service ethic of libraries can also be applied to scholarly publishing in libraries. When doing so, it informs the development and support of content dissemination in new and interesting ways that primarily support openness rather than cost recovery. Commercial publishers are reliant on serving their shareholders, not content users. Saarti and Tuominen sum this up well when they wrote: "Scholarly interests of sharing collide with commercial interests of generating profits" [15].

In the instances where University presses and Libraries have merged, their differing approaches to financial resources and business models has been a source of tension and illustrates how emerging library publishers differ from all other types of publishers. Because nearly all types of publishers in the past have been expected to recover the majority of their costs (along with limited institutional subsidies in the case of society publishers and university presses), it is challenging to consider a publishing program that does not assume cost recovery as a necessity. Library publishing, however, when seen as an active library-supported collection development strategy, is presenting that challenging question to the scholarly community. Graham Stone, in his thoughtful article about "New University Presses" or NUPs, notes that "These new publishing ventures, often based in the library, have harnessed the changes in the digital landscape and the rise of the open access movement to allow them to publish scholarly works, such as journals and monographs." He goes on to say that "Furthermore, a business model based on scholarly communication rather than profitability, but working on a cost recovery model appears to be contradictory . . . The Institution/Funder-pays model is the more appropriate model" [16].

Conversations within libraries about philosophy, and the need for cost recovery are essential in the development of library publishing business plans.

## 4. Three Case Studies of Library Publishing Programs

Most library publishing programs do not make their business plan publicly available; however, some elements of a library publishing program's business plans are evident through the information publicly displayed on their website and in the LPC's annual directory. A brief examination of three diverse program, outlined in Table 2, illustrate similarities in principles, a variety in scope, a wide range of staffing, and differences in services offered at these institutions.

**Table 2.** Case Studies of Library Publishing Programs.

**Institution #1: University of Minnesota Libraries—Publishing Services**

https://www.lib.umn.edu/publishing/about
Operates separately from the University of Minnesota Press which is housed in a different administrative unit at the institution. Administratively separate from the institutional repository, data repository, & digital humanities. [1]

| Principles: | Scope & Eligibility: | Staffing & Financials: | Development & Production Services: | Public Business Plan: |
|---|---|---|---|---|
| Library involvement is critical to advancing transparent scholarly and academic publishing practices. UMN Libraries have a commitment to Open Access, scholar-led publishing where creators maintain copyrights. | Publishes journals, monographs, dynamic scholarly serials, and course materials. No APCs allowed. U of MN affiliates and scholarly societies may apply to publish content with the University Libraries. Proposals reviewed biannually. | Director; Publishing Services Librarian; Development & Technology Staff; Publishing Services Coordinator. (3.5 FTE Total). External vendors used for production tasks. Funding Sources: library operating budget (75%); library materials budget (25%). Other financial information not available. | Basic Services (hosting, preservation, etc.) offered without charge to affiliates. Hosting charges apply to society-owned publications. Production (e.g., copy editing, typesetting, graphic design, etc.) and development charges apply to all publications. | Not Available. |

**Institution #2: University of Michigan Libraries—Michigan Publishing Services**

https://www.publishing.umich.edu/services/
Operates within the same office as the University of Michigan Press. The Press reports up administratively to the library and functions as a traditional university press. Also administered in the same office as the institutional repository. [1]

| Principles: | Scope & Eligibility: | Staffing & Financials: | Development & Production Services: | Public Business Plan: |
|---|---|---|---|---|
| MI Publishing Services staff are experts in scholarly publishing and "help increase the visibility, reach, and impact of scholarship." Emphasis on open access formats that advocate for author rights through new digital publishing models to ensure wider knowledge sharing. | Publishes books, journals, conference proceedings, digital projects, and course materials in print and electronic forms. Focus: Support for University of Michigan affiliates. | Publishing Services Director; Publishing Services Librarian; Publishing Services Coordinators; Community Manager (7 FTE Total). University Press and external vendors are used when needed. Funding Sources: library operating budget (50%); sales and hosting revenue (30%); charge backs (20%). Other financial information not available. | Full suite of services offered including: hosting, editing, typesetting, design, formatting (e.g., pdf, epub, OCR, etc.), digitization, web design, preservation, print on demand, Charges apply to most services. | Not available. |

**Institution #3: University of Pittsburgh Library System E-Journal Publishing**

https://www.library.pitt.edu/e-journals
Operates separately from the University of Pittsburgh Press which is housed in a different administrative unit at the institution. [1]

| Principles: | Scope & Eligibility: | Staffing & Financials: | Development & Production Services: | Public Business Plan: |
|---|---|---|---|---|
| Committed to helping research communities share knowledge and ideas through Open Access electronic publishing. They subsidize the costs of electronic publishing so that their "partners can focus on editorial content and scholarly collaboration". | Publishes Open Access eJournals. APCs allowed but no journals currently charge them. Focus: Publications that have: rigorous peer-review; an internationally recognized editorial board; a robust staff; and publish selectively from an open call for papers. No U of Pittsburgh affiliation required. | Director, Digital Repository Manager, Electronic Publications Manager, Library Specialists (4 FTE) Funding sources: library operating budget (75%); charge backs (25%). Other financial information not available. | Design services, assignment of standard identifiers, social media connections, analytics, consultations on editorial and management, indexing, archiving and preservation. | Not available. |

[1] Description of the library publishing program's principles, scope, eligibility, staffing financials, and services based off of a program's listed website.

## 5. Creating a Business Plan to Library Publishing

There has not yet been analysis or work done to define business plans for library publishing programs. This article uses the definition of business plan developed by Collier in 2005, and used in his 2010 edited volume, Business Planning for Digital Libraries: International Approaches:

> Business planning for digital libraries is here defined as the process by which the business aims, products and services of the eventual system are specified, together with how the digital library service will contribute to the overall business and mission of the host organizations. These provide the context and rationale, which is then combined with normal business plan elements such as technical solution, investment, income, expenditure, projected benefits or returns, marketing, risk analysis, management, and governance [17].

The anatomy of a library publishing business plan closely mirrors a template for a traditional, stand-alone business. However, because a library publishing program is nested within a larger organization, the financial section varies based on a university's budget model (discussed in Section 3) and their library's approach to funding these services. The authors of this paper recommend that libraries first identify the university's current budget model prior to writing a library publishing business plan.

The basic template for a library publishing business plan includes the following sections:

- Principles of Service
- Scope of Service
- Staffing and Governance
- Development & Production
- Financials
- Measures of Success

It is important to note, that if the institutional context calls for it, additional sections can be added to the business plan to strengthen alignment. This is especially true for libraries that are venturing into library publishing on an experimental basis—and for libraries that are in the process of advocating for the formalization of a publishing program. Useful additional sections for libraries in these positions include a PEST analysis (political, economic, social, technological) and a SWOT analysis (strengths, weaknesses, opportunities, threats). These sections can further illustrate the rationale behind the development of a publishing program [18].

The template used in this paper does not include a section on technology. Publishing technologies, specifically open source publishing technologies, are constantly growing in number and functionality. The authors highly recommend conducting a review of available publishing platforms. The Library Publishing Coalition offers members and non-members a number of resources on available technologies. (https://librarypublishing.org/)

The finalized business plan should be inclusive and detailed enough that administrators and campus partners can reference the plan and understand the functions and goals of the publishing program. The business plan can also act as a reference when questions arise from clients about the viability and sustainability of a new service. The ability to communicate the structure of and financial commitments of the publishing program is essential to conveying stability, knowledge of process, and boundaries. With the exception of the principles of service, it is expected that the business plan will need additional updates as staffing changes, library priorities shift, and as the program matures and grows.

### 5.1. Principles of Service

A library publishing business plan is a roadmap for the service. It explains to internal and external partners the details of how the program will travel from point A to point B. Principles of service,

in turn, explain to partners why the program is traveling at all. This is the intrinsic lead-in to a library publishing business plan. Principles can touch on themes mentioned earlier in this article, including: transparency, openness, and institutional support.

As a department or service offering of the library, library publishing programs inherit established mission statements, goals, and other strategic planning objectives from the library, and in turn the University. Although these objectives may convey the spirit of the service, a library publishing program will benefit from principles of service that are specific to the program. Developing and adopting principles of service will clearly define a library publishing program, communicate the program's purpose, and create a shared expectation of goals and outcomes.

Unlike an annual or strategic plan, principles of service should remain true given the, often unpredictable, ebbs and flows of passing years. Principles of service can be considered the "core" of the program and should not depend on a specific project or specific person. Principles should clear, accessible, and easy to share with clients and partners. Libraries with suites of scholarly communication services can leverage principles of service to help distinguish publishing services from other services offered within the organization. Drafting principles with library colleagues, including perspectives from digital humanities, copyright, and administration, allow for language that works in harmony among other services.

*5.2. Scope of Service*

One of the most challenging sections of the business plan, and the section most likely to change as the service is updated, is the scope of service. This section should address specific services that the library publishing program will provide, it could also highlight related services that the program will not provide. (For example, the library publishing program will not manage the inventory of print publications.) Additionally, this section is the section that will likely have the most dependencies with other sections. For established programs, this section will likely be a formal write-up of currently provided services within the program. For newly developed programs, this section should include the services that the program is ready to offer, and exclude services that the program hopes to provide in the future. Generally, this section should address the following questions:

- What type of publications will be published?
- Which authors/editors are eligible?
- What level of service will be provided to each publication?

Each of these questions requires a deeper consideration based on selected technologies, availability of staffing/personnel, and cost.

The most common types of publications published by library publishing programs are journals, monographs, and textbooks. However, as digital publishing tools grow, and the definition of scholarship broadens, programs may become publishers of increasingly difficult to categorize modes of scholarship. No matter the breadth of publication types, libraries should consider:

- What technologies will be needed to host and produce each type of publication?
- Are there other library or campus programs that currently serve the needs of the identified publication type?
- Will publishing staff be available to assist the editors of publications on an on-going basis (serials) or for only a limited time (monograph)?
- What is the average cost associated with each type of publication? Are these one-time costs or on-going?

Identifying the type of eligible clients for the publishing program will help the library build a customer profile for a marketing base. Even though the program may not be "selling" the final outputs, identifying who the service is for, will help communicate the program's principles of service to the appropriate audience. In specifying the programs' eligible clients, libraries should further consider:

- Does the library's mission focus on serving affiliated users?
- Does the program have a discipline specialty or focus?
- Can the program's selected technology work with affiliates and non-affiliates? Or are there EZproxy or Shibboleth requirements?
- Can the library and/or university budget cover expenses of non-affiliates?
- Will the program prioritize the works of different groups? (e.g., faculty, graduate students, undergraduates)

Across all the above mentioned points, is the question of what level of service the program will provide. This may be one of the harder questions to answer for a program that is just developing. However, once one publication is published, a program can run a project post-mortem to help identify how the skill sets of the individuals staffing the publishing program was leveraged and how much time went into the publication. Similarly, this question can also be answered throughout the initial publishing technology review—what processes can be automated using the available technology? (e.g., assigning DOIs, creating article metadata, password resets for platform users.) Generally, all of the following points should be considered:

- What can the technology for each type of publication automate?
- Do all publication types require the same amount of time and attention from the program staff?
- What will the editors of each publication be responsible for? What will the publisher be responsible for?
- How will clients contact the publisher?
- How will customer service be approached in relation to existing library services?

*5.3. Staffing*

Staffing within library publishing programs vary greatly. The 2018 Library Publishing Coalition Directory includes listings for programs with 0.25% of a full-time professional staff, all the way up to 16 full-time professional staff [2]. As noted in the previous section, the availability of staff directly impacts the services that a program can provide. A library can anticipate that this section of the business plan is inseparable to the program's Scope of Services.

Libraries drafting this section of the business plan should also consider where the publishing program is organizationally situated within the library. Since publishing may be a cross-departmental or cross-divisional effort, it is important to clearly describe where the program sits within the organization. Including this description for brand new programs will help colleagues throughout the library understand the reporting structure of the program.

Publishing programs need to define roles and responsibilities for each element identified in the Scope of Services. Programs that depend on the labor and/or time of library staff members in other units or departments, can formalize these relationships in the business plan in order to solidify cross-library buy-in. Although each element in the Scope of Services should be addressed in this section, the business plan is not an internal workflow document, so responsibilities may be identified at a general level and individual staff members may be identified by position, rather than name. These responsibilities will likely include:

- Technology development and support
- Marketing of services and recruitment of publications
- Production and development of publications, including additional processes identified in the Development & Production section
- Assessment, discovery, and promotion of individual publications
- Long-term strategic planning and goal setting (at publishing program level)

In addition to staffing, library publishing programs may find it beneficial to implement a governance structure. Unlike the day-to-day operations of the program, a governance structure

can provide recommendations to enhance the quality and future viability of the program. Building in the development of a governance structure can be a way to incorporate disciplinary faculty and other university stakeholders into the publishing program.

*5.4. Development & Production*

A variety of policies are required in order to make a library publishing program successful and sustainable. Policies guide decision making and can be referred to by administration or clients when questions arise. The need for policies is best summarized in *The Handbook of Journal Publishing* as policies address "what is to be published, how and why" [19]. Although an individual library publishing program may have policies unique to the program's goals and needs, there are a handful of policies that are essential to any publishing program.

5.4.1. Accepting Publications

Whether a publishing program anticipates publishing 1 or 100 publications a year, the program needs to consider how publications will be received by the library publisher. Many publishers use a call for proposals (CFPs) to solicit publications. Using a CFP, even if the respondents are few, enables publishers to advertise their service, while giving guidelines as to what will be accepted. Even for library publishing programs that are experimental, and willing to publish content with limited traditional publishing options, each program will likely have some limitations—especially involving staffing and technology. For library publishing programs just getting off the ground, and unsure of limitations, consider a CFP with open ended questions, this will enable submitters to describe their project without limiting answers to checkboxes.

Once proposals are submitted, each publishing program will need to determine how proposals are accepted or rejected. Again, the library publisher will want to consider which proposals are actually doable based on staffing and technology. There will likely be publications and projects that are just not possible given the program's available support. For proposals that are viable, each program will need to determine who gets to say "yes" and "no" to publications. This can be done by the staff working in the program, by a committee established by the program, or by library administration.

After a proposal is accepted, the library publishing program will need to develop an MOA (memorandum of agreement) or MOU (memorandum of understanding) for each publication. An MOA/MOU will clearly layout the expectations from each party and can include any necessary legal agreements or policies that are relevant to the relationship between publisher and publication. For libraries not familiar with MOA/MOU, consult the institution's office of general council or contract office.

5.4.2. Rights

Library publishers need clear statements about rights related to each publication. Policies may vary across individual publications, but the publishing program should create policies that address the following:

- Who does the copyright of a publication belong to?
- Who does the title of the journal belong to? (Could an editorial board member find a new publisher and move the journal/book series/conference proceeding?
- How can the content be used? (This question can be addressed by the addition of a Creative Commons license.)
- How can either party end the business relationship between publisher and publication?

Individual publications, especially those with multiple authors, will need to create publication-specific policies to ensure that content within the publication is following copyright and/or licensing policies. As a publisher, it is important to assist editors or editorial boards that are

new, or those that have questions related to rights. Set up formal channels of communication and encourage publication editors to reach out for support.

### 5.4.3. Privacy

User privacy statements need to be included on each digital publication or digital publication access point. Chances are that the publishing program's selected software, especially if using a hosted solution, will include a privacy policy. Make sure that staff working on publications understand the privacy policies and are able to communicate the policies to users of the platform. For publications that require registration for readers, authors, or reviewers, make sure that any default privacy statements are correct and that all users are prompted to read the privacy/user agreement before entering any information into the system.

### 5.4.4. Distribution & Marketing Policies

Because the majority of library publishers publish content that is openly accessible, publishing programs will need to have unique marketing and distribution tactics not as common among traditional publishers and university presses. Setting distribution and marketing policies will clarify expectations between authors/editors and the publisher. If the publishing program sells print copies of books, will there be a markup fee? Can the author, as the copyright holder, set up their own digital storefront? Even in the world of open access publishing there is a need for policies related to distribution. A library publisher with the staff time and expertise may want to be the party responsible for applying to databases and indexes for each publication. Additionally, the publisher can take the lead on advertising or marketing publications. This may be something that the author/editor does not think of, especially if the publication is available online for free, however, the publisher will want to see a publication attract as many readers as it can. It is never too soon to work with editors/authors to develop a strategy for distribution and marketing, having a policy in place when a potential publication reaches the library publishing program will make any effort much more successful.

### 5.4.5. Preservation Policies

Preservation of library published content continues to be an area under investigation. In 2017, the Library Publishing Coalition noted that programs are "making slow but thoughtful progress on digital preservation" [2]. Although libraries continue to improve policies around the preservation of library published content, there are a number of approaches that can be taken to ensure that published works are preserved. Public Knowledge Project (PKP) and bepress, common library publishing platforms, allow users to set up accounts through Global CLOCKSS program (Controlled Lots of Copies Keeps Stuff Safe from Stanford University). Additionally, PKP offers a private preservation network available to platform users who are unable to join the Global CLOCKSS program. Portico is also an option for library publishers, and is the most common journal and ebook preservation tool used by libraries to preserve purchased content. Portico requires membership with fees based on journal or ebook revenue [20].

Regardless of whether or not a library publishing program is connected with preservation tools, a library publishing program should develop a clear policy that can address author/editor questions about both short- and long-term preservation. The policy should also address *what* content is to be preserved. Additionally, programs will want to consider:

- Will the publishing program preserve all publications?
- What about publications that cease or move to another publisher?
- Will a journal's webpages be preserved, or just PDFs?
- Will production files be preserved, or just version of record?

Preservation will likely be a policy that requires the expertise of librarians beyond the publishing program. It is also a policy that will need updating as technologies and best practices change.

Editors and authors want a publisher that will look out for published content for the long term, a successful preservation policy should address this.

*5.5. Financials*

Unlike other scholarly communication services, publishing has well-documented, though debated, costs associated with the service [21]. Libraries are especially sensitive to costs set by publishers, therefore a library as publisher has the opportunity to be especially transparent and clear in the costs associated with publishing. The development of the financials section of the business plan will need to be done in close consultation with library administration, it is likely that library has pre-developed language and/or templates for communicating costs. The basic financial structure of the program will likely be addressed in earlier sections of the business plan, however, the financials section should address the following questions:

- How will the service fit into the library's budget model?
- How can/will the service leverage the university's budget model?

  o    Will staffing and core technologies be paid for by the library's budget or covered by publishing revenue?
  o    Will the service charge fees for any/all services?
  o    How will service rates be calculated?
  o    What expenses will potential revenue cover?

- Which expenditures are flat versus usage-based?
- Which pre-existing memberships or technologies will the program use?
- How will costs, charged directly to clients or covered by the library, be communicated to clients?

Like earlier sections, the financials section requires that libraries estimate growth of the program in order to calculate costs. In addition to staffing and core technologies (digital publishing platforms), libraries need to consider expenses that fluctuate based on volume. Some of these costs may be:

- Identifiers (DOIs, ISSNs, ISBNs)
- Graphic design for individual publications
- Material for marketing and promotion
- Licenses for production tools (InDesign, iThenticate, Overleaf)
- Memberships for preservation and publishing best practices (Portico, COPE, etc.)

Additionally, each individual title should also have a budget assigned to it. The program's approach to publication level planning should be included in the financials section, this can be done by including a template or spreadsheet that is used to structure the relationship between author/editor and publisher. Being able to express to authors what resources are needed to launch and maintain their publication helps communicate expectations and outlines where they need to partner to provide additional resources for elements or features that are not currently supported by the service.

*5.6. Measures of Success*

Given the often experimental nature of library publishing, and the lack of longitudinal studies on library publishing, determining measures of success for a library publishing program can be a challenge. Measures of success will be determined based on each publishing program's principles of service and the parent institution's mission and vision.

To do this, Publishing programs may find measures of success tied to individual publications and projects. Measures of success for individual publications, especially those available free of cost, and therefore not being measured based on revenue, frequently fall into three general areas:

- Sustainability: Is the publication able to recruit reviewers, editors, and authors? Is the publication meeting publication-specific goals?
- Scalability: Is the publication able to respond to increased readership? Are editorial workflows keeping up with an increase in content?
- Visibility: Is the publication attracting readership? Is the publication being cited? When eligible, is the publication included in disciplinary-appropriate indexes?

However, the diversity of library publishing portfolios means that measures of success do not always work when tied to specific publications, especially books and other non-serials, whose content is not likely to grow over time. Measures of success for the overall publishing program " . . . must also be able to demonstrate that they are fulfilling the traditional roles of scholarly publishers" [19]. Some library publishers have principles of service that may vary drastically from "traditional publishers," making it important for a successful publishing program to also meet the needs requested by their clients. With that in mind, the same measures of success used to evaluate individual publications can be used to measure the success of the overall publishing program:

- Sustainability: Is selected technology still meeting publication needs? Are publishing staff able to maintain developed workflows?
- Scalability: Is there a growth in number of publications? Are additional services being added as requested?
- Visibility: Is there campus awareness of the publishing program?

Additionally, staff in library publishing should be aware of other measures of success that are used across library services. If a publishing program has services that include outreach and education, consider meeting with colleagues in library information literacy units to determine appropriate evaluation metrics for publishing services that extend beyond publications. Measures of success is another section of a library publishing business plan that can benefit greatly from vertical alignment with a library's related services and units.

## 6. Conclusions

In response to the variety of issues in scholarly communication, the development of library publishing programs is one way libraries have become active participants in the growing open access publishing landscape. Business plans for library services, especially for scholarly communication services, are not yet commonplace. However, by creating and adopting a business plan for library publishing programs, libraries can formalize a relatively new service within the unique structures of academic libraries. A library publishing business plan will provide a clear understanding of the program's goals and services, and will provide a path for growth and assessment in the long and short term. Its development offers the opportunity for the library's leadership and staff to discuss and create framing principles, which provide a foundation for communicating the goals and purpose of the service. The remaining elements of a robust business plan provide a structure for a program's operations and clear communication.

**Author Contributions:** Conceptualization, E.M. and K.M.; Introduction, E.M. and K.M.; Background and Development, K.M.; Anatomy, E.M. and K.M.; Conclusion, K.M.; Writing-Original Draft Preparation, K.M. and E.M.

**Funding:** This research received no external funding.

**Acknowledgments:** The authors thank Shaan Hamilton for his feedback on Section 3: Institutional Budget Models and Their Impacts.

**Conflicts of Interest:** The authors declare no conflict of interest.

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
