# Peer review of "Developing a Business Plan for a Library Publishing Program"

_publications, doi:10.3390/publications6040042_

Round 1

Reviewer 1 Report

I think that this topic would be of interest to Publications readers. It seems to address both a growing area of interest in the field and a gap in the literature.

The authors do a good job of laying out the components of a business plan, and the article is well organized and coherent.

However, there are a number of issues that I think need to be addressed before the article would be ready for publication, and these areas include both content and writing.

In terms of content:

The introduction would benefit from the addition of a single paragraph that clearly states the purpose of the article, how that purpose will be addressed, and why the topic is important.  The authors provide a good overview of the history and current state of library publishing, but it is not until the very end of the introduction that the idea of business plans are introduced and, even then, it is not clear how they will be addressed-- for instance, is this a research article with content analysis of existing business plans?  A "how we did it" article describing one institution's experience? Or more of a general guide?  And, in any of these cases, why is this topic important and how will it benefit the reader? Further, because this isn't a research article, I think it would help if at the end of the introduction the authors included a brief guide to the rest of the article- what issues/topics/questions will be tackled and in what order? As I said, this is important since this isn’t a research article per se, so it doesn’t follow the typical intro, lit review, findings, discussion, conclusion format.

I also think the article would benefit if the authors briefly but more clearly defined some of the models of library publishing-- throughout the article they repeatedly mention that there are many different models and that library publishing can focus on books, journals, student publications, individual works-- all of these, some combination of these, etc.  I think if a framework of the potential models was laid out, it would help to better frame the paper and also clarify the relationship between library publishing and other areas referenced throughout the article such as open access, scholarly communications and institutional repositories. 

Section 2 about the Minnesota Library Publishing Program seems superfluous.  The rest of the article does not seem to use the Minnesota model as a reference point, or draw on examples from that program, so it doesn't seem necessary to include that background.  If article included examples from Minnesota or reflections on their process and what worked or didn't, it would make sense, but as it is the rest of the article just gives very general advice.

The overview of the components of the business plan varies in level of depth, detail, and clarity.  For instance, sections 3.4.1 on accepting publications and 3.4.5 on preservation are good examples of sufficient depth and detail.  Some other sections could use more elaboration and specificity.  For example, 

Paragraph 3.3.3 Planners and Administrators could be expanded—what kinds of administration models are available? How do they work? Which administration model is best suited to which service model?

Similarly, 3.3.4—in describing how scholars might be recruited and contribute to the publishing, the author states “How they are asked to provide this knowledge can vary” … give some examples.

In several places throughout the description of the business plan, the authors state that aspects of the business plan, policies, etc., will depend on the publishing model… however, they never clearly lay out the various models available. This gets back to my earlier comment about laying out a framework early on-- then the authors could refer back to that framework and provide specific examples for different models.

The authors should remember that they are trying to provide advice and a model for people who might not have much experience with business plans or even with library publishing, so a certain level of detail and guidance is needed.

In terms of writing, the paper needs a thorough proofread and copy edit.  

There are numerous typos, misspellings, and small grammatical mistakes throughout.

A few specific notes:

In the introduction the authors state that ARL has 123 members in intro but 124 later in the manuscript.

Line 557 include citation or footnote for Stone

Most distressingly, Paragraphs 3.2 and 3.2.1 are exactly the same—only the headers differ.  The paragraph makes sense for 3.2 Scope of Services.  3.2.1 is supposed to address types of publications, but instead repeats the paragraph for scope of services.  The first sentence in paragraph 3.2.2. eligibility repeats the same content again.  This was a major error, and it is hard to judge the overall quality of the manuscript highly when an entire section is missing.

Author Response

As attached. 

Reviewer 2 Report

Dear Authors, thank you for putting this together. This document is a fine start to anyone looking to develop a business plan for library publishing. I particularly like that you highlight the funding models for libraries and how that intersects with the mission - and, inevitably, how that will intersect with the business plans. With this in mind, I have a few suggestions to make this document stronger. 

First, are there sample / model business plans that you can highlight out there? This could include your own. These may not be openly available, but they may be in some cases. 

Second, in some sections you give extensive recommendations of things to consider, while others are a bit thin. I have some specific comments about this to fill out the document as a whole, listed below: 

Lines 235-246 about budget models of universities - you go into details about one at your institution. At least mentioning some other budget models that fund libraries would be an excellent addition here to get readers on the right path to thinking about what their model might be and how it would look different from the one mentioned and described so well here. 

Lines 313-325 - section 3.2.1 "Types of Publications" - in my review copy, the text of this section was exactly the same as the text in the introduction, section 3.2, lines 299-311. I fear a copy/paste error. 

Lines 327-340 - section 3.2.2 "Eligibility" - the model where no affiliation or association is needed for a journal to publish with a university library is not mentioned here. (This is obviously a self-centered mention, since this is the model that this reviewer's library publishing program operates under.) In this model, while an affiliation with the university is not needed to publish with them, affiliates get certain benefits such as a discount on services, but might get other benefits such as in-person training and assistance. 

Lines 358-364 - section 3.2.4 "Quality" - the text in this section deals in a large part with selection criteria rather than quality assessments that we might see associated with journals or monographs. I recommend renaming this to selection criteria, or giving more information about how a business plan might be concerned with quality of the publication. 

Line 368 - under Section 3.2.5 "Service Menu" - it seems that Digital Commons and Public Knowledge Project are equated here as "somewhat inexpensive". I think this distinction deserves to be teased out more here. There are "free" options (open source) available like PKP's OJS, but they are not free to implement as they require expertise and infrastructure that may not already be present. Meanwhile there are hosted services like Digital Commons and PKP's Publishing Services branch (which you may have meant to refer to above) which take some of the implementation costs out of the picture and replace it with a fee paid to the company. This could factor in to your staffing & governance section later as well. In fact these two sections (Staffing & Governance and the Service Menu sections) are very closely entangled because what you have the capability for in your library already will influence what you are able to provide easily and what you may need to outsource. Making this point a bit more forcefully would help here. 

Lines 422-430 are very vague about the way the library publisher interacts with scholars. It might be best to give more details about what this might look like. For example, in our journal publishing programs, the editors of the journals assemble the editorial boards and the peer reviewers from their networks, so the library really does not have to do much. However, in monographs the library may have to solicit reviewers. 

Lines 461-473 - section 3.4.2 "Rights" - this is a place where some more suggestions could be helpful. It's a chance to advocate for embracing the most open policies as well. At the very least, some of the COPE (Committee on Publication Ethics) best practices and the OASPA Code of Conduct could be referenced here as places for people to look for inspiration and guidance. 

Lines 498-521 - section 3.4.5 - this is an excellent example of a section that provides lots of references and guidance questions. I would like to see many of the sections mentioned above look more like this one. 

To close, I want to re-emphasize how important this document is and how helpful it could be to many library publishers. My comments here are intended to help bolster it to be the best it can possibly be and contain as much useful information from many contexts as possible. I look forward to seeing this (after revisions) in print and sharing it with colleagues. 

Author Response

As attached. 

Reviewer 3 Report

This is a solid practical paper on the relevancy and structure of a business plan for a library publishing program in a university library setting. The paper is mostly US centered although similar developments are also taking place elsewhere, for example in European LERU universities (UCL Press, Helsinki University Press). However, the rationale behind OA publishing programs is well presented. This is a very useful paper, for example, for those organizations wishing to build their own OA Presses.

As a reader, I am wondering why are libraries especially suitable for running university presses? Of course, they have the motivation (because of serial crisis) and are heavily committed to the general OA movement. However, should these same topics/questions should be answered by other potential actors running OA presses as well?

Would it be possible to attach a real or imaginary business plan of an OA press in the paper?

Author Response

As attached.

Round 2

Reviewer 1 Report

This revision is very well done, and I recommend publication.  All of my concerns have been addressed.